# Saporin Toxin Delivered by Engineered Colloidal Nanoparticles Is Strongly Effective against Cancer Cells

**DOI:** 10.3390/pharmaceutics14071517

**Published:** 2022-07-21

**Authors:** Lucia Salvioni, Filippo Testa, Linda Barbieri, Marco Giustra, Jessica Armida Bertolini, Giulia Tomaino, Paolo Tortora, Davide Prosperi, Miriam Colombo

**Affiliations:** Department of Biotechnology and Bioscience, University of Milano-Bicocca, Piazza della Scienza 2, 20126 Milano, Italy; lucia.salvioni@unimib.it (L.S.); f.testa@campus.unimib.it (F.T.); l.barbieri17@campus.unimib.it (L.B.); m.giustra2@campus.unimib.it (M.G.); jessica.bertolini@unimib.it (J.A.B.); g.tomaino3@campus.unimib.it (G.T.); paolo.tortora@unimib.it (P.T.); davide.prosperi@unimib.it (D.P.)

**Keywords:** ribosome-inactivating proteins (RIPs), toxins, Saporin, colloidal nanoparticles, cancer therapy, breast cancer

## Abstract

Ribosome-inactivating proteins, including Saporin toxin, have found application in the search for innovative alternative cancer therapies to conventional chemo- and radiotherapy. Saporin’s main mechanism of action involves the inhibition of cytoplasmic protein synthesis. Its strong theoretical efficacy is counterbalanced by negligible cell uptake and diffusion into the cytosol. In this work, we demonstrate that by immobilizing Saporin on iron oxide nanoparticles coated with an amphiphilic polymer, which promotes nanoconjugate endosomal escape, a strong cytotoxic effect mediated by ribosomal functional inactivation can be achieved. Cancer cell death was mediated by apoptosis dependent on nanoparticle concentration but independent of surface ligand density. The cytotoxic activity of Saporin-conjugated colloidal nanoparticles proved to be selective against three different cancer cell lines in comparison with healthy fibroblasts.

## 1. Introduction

In the research into new drugs for tumor treatment, ribosome-inactivating proteins (RIPs) have recently emerged as promising agents of interest because of their potent cell-cycle-independent activity against cancer cells. RIPs are especially advantageous due to their enhanced thermostability, structural solidity against denaturation or alteration caused by proteolytic enzymes, and ease of handling [1,2].

RIP activity is traditionally attributed to rRNA N-glycosylase; this specific depurination leads to a permanent inactivation of the ribosome, irreversibly blocking protein synthesis and inducing DNA damage. Consequently, all RIPs cause cell death by apoptosis at very low dosages (less than a few milligrams/kg) [3,4].

Among the RIPs toxin family, Saporin (Sap, from the plant *Saponaria officinalis*) has attracted increasing attention in the last two decades thanks to its low systemic toxicity, due to its low propensity to enter cells in the absence of a specific carrier [5].

In vivo results have demonstrated the efficacy of immunotoxins (ITs) obtained by the conjugation of Saporin to monoclonal antibodies, which represents the ideal toxic moiety in cancer patients who are refractory to traditional therapies, including surgery, radiation therapy and chemotherapy [6,7].

Nevertheless, despite all of these significant benefits, the application of Saporin technology in current clinical practice is restricted to hematological malignancies and limited by several hurdles that need to be overcome related to the problems of immunogenicity, vascular leak syndrome due to endothelial toxicity and inefficient intracellular release [7,8,9]. The latter represents the main limitation on Saporin antitumor activity due to the very low protein uptake efficiency of cancer cells and negligible endosomal escape once internalized [9,10].

One possible approach to overcome this issue is to exploit Saporin conjugation to a suitable nanocarrier that improves its intracellular release [11]. In our ongoing effort to study the impact of surface coating of colloidal nanoparticles to achieve high uptake efficiency and optimal endosomal escape, we have developed biocompatible iron oxide nanoparticles coated with a pH-sensitive amphiphilic polymer [12,13]. This multidentate polymer, termed PMDA, obtained by modification of poly(isobutylene-alt-maleic anhydride) with 75% dodecylamines, demonstrated excellent endosomal escape capability upon endosome maturation [14]. PMDA-coated iron oxide nanoparticles, termed MYTS, proved to be non-toxic both in vitro and in vivo [12,15] and very easy to functionalize with different bioactive molecules, including proteins [16,17,18,19].

In the present work, we functionalized MYTS with Sap for the purpose of improving the intracellular release of toxins in cancer cells. Sap was reversibly conjugated to PMDA through a disulphide bond, sensitive to the cytosolic reducing environment. We used SK-BR-3 as a breast cancer cell model and NIH-3T3 fibroblasts as healthy control cells.

## 2. Materials and Methods

### 2.1. Synthesis of MYTS-Sap-1, MYTS-Sap-2 and MYTS-Sap-3

#### 2.1.1. Chemicals and Materials

Twelve-nanometer iron oxide nanoparticles (NPs) were purchased from HiQ-Nano (Arnesano, Italy); 2,2′-(Ethylenedioxy)bis(ethylamine) (EDBE), Boric acid, CHCl_3_, Dodecylamine (99%), EtOH, Poly(isobutylene-alt-maleic anhydride) (PMA) Mw ~6000 and N-(3-Dimethylaminopropyl)-N′-ethylcarbodiimide hydrochloride (EDC) were purchased from Sigma-Aldrich (Saint Louis, MO, USA); and sulfosuccinimidyl-6-(3′-(2-pyridyldithio)propionamido)hexanoate (sulfo-LC-SPDP) was purchased from CovaChem (Loves Park, IL, USA). Water used in all procedures was purified by passing through a MilliQ Millipore (Burlington, MA, USA) system.

#### 2.1.2. Phase Transfer of Iron Oxide Cores

Toluene was removed from the iron oxide NPs using a Laborota 4000 efficient Rotary Evaporator (Heidolph Instrument, Schwabach, Germany) and NPs were resuspended in 2 mL CHCl_3_ and quantified by UV–Vis spectroscopy, as previously described [20]. Polymer coating and phase transfer of iron oxide cores was performed using PMDA at a polymer/NP ratio of 150/nm^2^. The synthesis of PMDA polymers and the ratio used were optimized in previous publications [21,22]. Specifically, for 2 mL of NPs 0.037 M, 347.2 µL of 0.5 M PMDA were added. The solvent was evaporated under reduced pressure and the obtained NPs were dispersed by sonication in aqueous 0.51 M Saline Borate Buffer (SBB), pH 9.0, for 20 min. The solution was then diluted in sterile MilliQ and washed by centrifugal filtration using an Amicon^®^ Ultra centrifugal filter (100 kDa) (Merck Millipore, Bulington, MA, USA). The PMDA-coated NP (MYTS-PMDA) content was quantified as described above. All the synthetic procedures reported here and in the following paragraphs were conducted under sterile conditions.

#### 2.1.3. EDBE Functionalization of MYTS-PMDA

A volume of 8 µL 0.1 M EDBE and a volume of 6 µL 0.05 M EDC were added to 10 mg of MYTS-PMDA (10 mg/mL). The mix was kept under stirring for 2 h at RT. Byproducts and the unconjugated EDBE were removed with an Amicon^®^ Ultra centrifugal filter (50 kDa) using sterile MilliQ water. The final concentration of EDBE-conjugated MYTS (MYTS-EDBE) was quantified as described above.

#### 2.1.4. Sulfo-LC-SPDP Functionalization of MYTS-EDBE

A volume of 506.6 µL 0.019 M sulfo-LC-SPDP was added to 10 mg of MYTS-EDBE (10 mg/mL). The mix was kept under stirring overnight at 4 °C. Byproducts and the unconjugated sulfo-LC-SPDP were removed with an Amicon^®^ Ultra centrifugal filter (50 kDa) using sterile MilliQ water. The final concentration of sulfo-LC-SPDP-conjugated MYTS (MYTS-SPDP) was quantified as described above.

#### 2.1.5. Saporin Conjugation

To achieve a specific conjugation to MYTS-SPDP through the formation of a disulfide bond, a recombinant version of Sap was produced, with the addition of a unique Cys residue located at the C-terminus of the protein (C-Sap; details about protein production are reported in the Appendix A). However, the Cys residue could lead to the dimerization of the protein, making it unavailable for conjugation to the -SH group of MYTS-SPDP. To obtain monomeric Saporin with a free -SH group available for conjugation to MYTS-SPDP, the protein was activated with 1 mM EDTA and 50 mM DTT for 30 min at RT. The excess of DTT and EDTA were removed with a Zeba™ Spin desalting column (7 kDa) (Thermo Fisher Scientific, Waltham, MA, USA), using 50 mM potassium phosphate, pH 7.2, and 150 mM NaCl as final buffer.

MYTS-Sap-1, MYTS-Sap-2 and MYTS-Sap-3 were synthetized, using, respectively, 50, 100 and 200 µg of reacted Saporin per mg of MYTS-SPDP. The mixtures were kept under stirring for 18 h at 4 °C. Unreacted protein and byproducts were removed with an Amicon^®^ Ultra centrifugal filter (100 kDa) using 50 mM potassium phosphate, pH 7.2, and 150 mM NaCl as washing buffer.

#### 2.1.6. Saporin-Alexa Fluor™-647 Labelling

A Zeba™ Spin desalting column (7 kDa) was used to exchange Saporin buffer to 0.09 M NaHCO_3_. Then, a volume of 11.65 µL Alexa Fluor™-647-NHS-ester (5 mg/mL in DMSO) (Thermo Fisher Scientific, Waltham, MA, USA) was added to 600 µg of Saporin (2 mg/mL), to obtain a molar dye/protein ratio of 2.5, and kept under stirring for 2 h at RT. At the end of the incubation, the unreacted dye was removed with 2 further washes with a Zeba™ Spin desalting column (7 kDa), using 50 mM potassium phosphate, pH 7.2, and 150 mM NaCl as final buffer.

The degree of labelling was calculated as dye concentration (M)/protein concentration (M), based on UV–Vis spectroscopy; 280 nm and 647 nm absorbances were used, respectively, for the quantification of the protein and the dye, using the following formulas that subtract the contribution of the dye at 280 nm:(1)$$\mathrm{Degree}\;\mathrm{of}\;\mathrm{labelling}\;=\frac{{\mathrm{Abs}}_{647\;\mathrm{nm}}}{\varepsilon_{\mathrm{Alexa}647}\times l\times \mathrm{protein}\;\mathrm{concentration}\;\left(M\right)}$$
where ε_Alexa647_ (molar extinction coefficient of Alexa Fluor™-647-NHS-ester) is 239,000 cm^−1^ M^−1^, l (optical path length) is 1 cm and the protein concentration (M) was calculated as:(2)$$\mathrm{Protein}\;\mathrm{concentration}\;\left(M\right)=\frac{{\mathrm{Abs}}_{280\;\mathrm{nm}}-0.03\times {\mathrm{Abs}}_{647\;\mathrm{nm}}}{\varepsilon_{\mathrm{Sap}}\times l}$$
where 0.03 is the correction factor recommended by the supplier to remove the contribution of Alexa Fluor™-647-NHS-ester from the 280 nm absorbance, l is 1 cm and ε_Sap_ is 23,380 (calculated by ProtParam [23]).

### 2.2. NP Characterization

#### 2.2.1. Transmission Electron Microscopy (TEM)

The NP suspension was deposited onto a formvar-coated 200-mesh copper grid (Ted Pella, Redding, CA, USA) and allowed to dry before examination. NPs were then visualized with a Jeol JEM 2100Plus (Jeol, Tokyo, Japan) electron microscope equipped with a 9 MP complementary metal oxide superconductor (CMOS) Gatan Rio9 digital camera (Gatan Inc., Pleasanton, CA, USA). The core diameter (*n* = 100) was calculated by processing the recorded images with Fiji/ImageJ software [24].

#### 2.2.2. Dynamic Light Scattering (DLS) and ζ-Potential Analysis

NP hydrodynamic diameter and ζ-potential were analyzed on a Zetasizer Nano ZS ZEN3600 (Malvern Panalytical, Malvern, Worcestershire, UK), operating at a light source wavelength of 633 nm and a fixed scattering angle of 173°. For both DLS and ζ-potential analysis, the purified samples were diluted in distilled water to a concentration of 30 µg/mL and briefly sonicated prior to the analysis. The results were expressed as means ± standard deviations (SDs) of three measurements.

#### 2.2.3. Conjugation Efficacy

The conjugation efficiency of Saporin was calculated after MYTS-Sap-Alexa-1, MYTS-Sap-Alexa-2 and MYTS-Sap-Alexa-3 synthesis as µg of conjugated Saporin/µg of reacted Saporin. The quantities (µg) of conjugated Saporin in the MYTS-treated samples were calculated via UV–Vis spectroscopy based on Alexa Fluor™-647-NHS-ester 647 nm absorbance and degree of protein labelling. Specifically, MYTS-SPDP nanoparticles were analyzed to calculate the Abs 550 nm/Abs 647 nm ratio. This ratio was used to determine the iron oxide core aspecific 647 nm contribution of MYTS-Sap-Alexa-1, MYTS-Sap-Alexa-2 and MYTS-Sap-Alexa-3. This value was subtracted from the 647 nm absorbance of each sample (Abs_647 sub_), and Saporin concentration (M) and conjugation efficiency (%) were calculated as follows:(3)$$\mathrm{Saporin}\;\mathrm{concentration}\;\left(M\right)=\frac{{\mathrm{Abs}}_{647\;\mathrm{nm}\;\mathrm{sub}}}{\varepsilon_{647\;\mathrm{nm}\;}\times l\;\times \mathrm{degree}\;\mathrm{of}\;\mathrm{labelling}}$$
where ε_Alexa647_ (molar extinction coefficient of Alexa Fluor™-647-NHS-ester) is 239,000 cm^−1^ M^−1^ and l is 1 cm;
(4)$$\mathrm{Conjugation}\;\mathrm{efficiency}\;\left(\%\right)=\frac{\mathrm{Saporin}\;\mathrm{concentration}\;\left(\mathrm{mM}\right)\times \mathrm{Protein}\;\mathrm{MW}\times \mathrm{Volume}\;\left(\mathrm{mL}\right)}{\mathrm{Reacted}\;\mathrm{Saporin}\;\left(µg\right)}$$
where Saporin MW is 34,753 Da (calculated by ProtParam).

### 2.3. Cell Culture

SK-BR-3 cells were cultured in Dulbecco’s Modified Eagle’s Medium (DMEM) High Glucose/Ham’s F12; U87, HeLa and NIH-3T3 cells were cultured in DMEM High Glucose. All media were supplemented with 2 mM L-glutamine, penicillin (50 U/mL), streptomycin (50 mg/mL) and 10% of either Fetal Bovine Serum (FBS) for SK-BR-3, HeLa and U87 or Calf Bovine Serum (CBS) for NIH-3T3. All cells were cultured at 37 °C in a humidified atmosphere containing 5% CO_2_ and sub-cultured prior to confluence using trypsin/EDTA. All the cell media and supplements were purchased from Euroclone (Pero, Italy).

### 2.4. Cell Viability Assay (MTT Assay)

SK-BR-3, NIH-3T3, U87 and HeLa cells were seeded on a 96-well dish at a density of 1 × 10^4^ (SK-BR-3 and U87) or 5 × 10^3^ (NIH-3T3 and HeLa) cells/well and grown for 24 h in the appropriate medium. Cells were then incubated with MYTS-SPDP, MYTS-Sap-1, MYTS-Sap-2 and MYTS-Sap-3 at different concentrations (range: 1–50 µg/mL) and free Saporin at either the maximum concentration found in the MYTS-treated samples (corresponding to that of MYTS-Sap-3 at the highest concentration) for HeLa, U87 and NIH-3T3 cells or 2×, 3× and 4× the concentration for SK-BR-3. The equivalent Saporin nM concentrations for each sample can be found in Appendix A. After 24 h, the medium was refreshed and cells were incubated for 4 h at 37 °C with 15 µL of 3-(4,5-dimethyl-2-thiazolyl)-2,5-diphenyl-2H-tetrazolium bromide (MTT) stock solution (Promega, Milano, Italy). At the end of the incubation, 0.1 mL of MTT solubilizing solution (Promega, Milano, Italy) was added to each well for 1 h at RT under shaking to solubilize the MTT formazan crystals. Absorbances were read with an EnSight™ Multimode Plate Reader (PerkinElmer, Waltham, MA, USA), using a test wavelength of 570 nm and a reference wavelength of 620 nm. Cellular viability was calculated by normalizing the treated sample absorbance against the absorbance of the untreated control sample. The results were expressed as the means ± standard deviations of four independent experiments.

### 2.5. Apoptosis Assay

SK-BR-3 and NIH-3T3 cells were seeded on a 12-well dish at a density of 2.5 × 10^5^ cells/well and grown for 24 h. The cells were incubated with MYTS-SPDP, MYTS-Sap-1, MYTS-Sap-2 and MYTS-Sap-3 at concentrations of 75 µg/mL, with the highest concentration of free Saporin being found in the MYTS-treated samples (MYTS-Sap-3, equal to approximately 166.6 nM). After 24 h, floating cells were collected in flow cytometry tubes and adherent cells were detached with trypsin and pooled together with corresponding floating cells. Cells were centrifuged and washed once with ice-cold PBS, then resuspended in 100 µL Annexin-binding buffer (10 mM HEPES, 140 mM NaCl and 2.5 mM CaCl_2_, pH 7.4) and incubated with 5 µL of Annexin V-Alexa Fluor™-488 solution (Thermo Fisher Scientific, Waltham, MA, USA) and 15 µg/mL Propidium Iodide (PI) for 15 min at RT in the dark. At the end of the incubation time, 200 µL of Annexin-binding buffer was added to the suspension and the cells were analyzed with a Gallios™ Flow Cytometer (Beckman Coulter Inc., Brea, CA, USA). For each analysis, 1 × 10^4^ events were acquired after gating on single cells, and a sample of untreated cells was used to set the appropriate gate for the regions of positivity. The percentages of positive cells were reported as the mean ± SD of three individual experiments for SK-BR-3 and of a single experiment for NIH-3T3. Untreated cells were used as controls. The data were analyzed using FlowLogic™ Software (Inivai Technologies, Melbourne, Australia).

### 2.6. Protein Synthesis Inhibition Assay

SK-BR-3 and NIH-3T3 cells were seeded on a black 96-well dish at a density of 1 × 10^4^ and 5 × 10^3^ cells/well, respectively, and grown for 24 h. The cells were incubated with MYTS-SPDP, MYTS-Sap-1, MYTS-Sap-2 and MYTS-Sap-3 at a concentration of 50 µg/mL and with the highest concentration of free Saporin being found in the MYTS-treated samples (MYTS-Sap-3, equal to approximately 111.1 nM). After 6 h, the inhibition of protein synthesis was evaluated using a Click-iT^®^ Plus OPP Alexa Fluor^®^-488 Protein Synthesis Assay Kit (Thermo Fisher Scientific, Waltham, MA, USA), according to the manufacturer’s instructions. At the end of the procedure, cell green fluorescence signals were imaged with a PAULA microscope (Leica Microsystem, Buffalo Grove, IL, USA) and the images were analyzed with ImageJ software. Specifically, for each cell line, the original images were converted to greyscale 8-bit images, an automatic threshold was generated using the untreated sample with the “Auto Threshold” function and the same intensity value was used to generate the threshold for all the treated samples. Then, the “Watershed” function was used to separate any cell clusters into single cells and the “Measurement” function was used to measure the mean grey value of the above-threshold cells. The means and standard deviations of all cells mean grey values for each image were calculated and normalized against those of the untreated sample.

### 2.7. Cellular Uptake Assay

SK-BR-3 and NIH-3T3 cells were seeded on a 12-well dish at a density of 2.5 × 10^5^ cells/well and grown for 24 h. The cells were incubated with MYTS-SPDP, MYTS-Sap-Alexa-1, MYTS-Sap-Alexa-2 and MYTS-Sap-Alexa-3 at an equal MYTS concentration of 10 µg/mL or at an equal Saporin concentration of 5.4 nM (found in MYTS-Sap-Alexa-1 10 µg/mL). The highest concentration of free Alexa-labelled Saporin used was found in the MYTS-treated samples, these being, respectively, 22.2 nM and 5.4 nM. After 4 h, the NP-containing medium was removed and cells were detached with trypsin, collected in flow cytometry tubes and washed twice with PBS. Cells were finally resuspended in 300 µL PBS 2 mM EDTA and analyzed using a Gallios Flow Cytometer (Beckman Coulter). For each analysis, 1 × 10^4^ events were acquired after gating on single cells, and the median fluorescence intensities of samples were normalized against a sample of untreated cells. The data on the normalized medians of fluorescence intensity were reported as means ± standard deviations of three independent experiments. Untreated cells were used as a control. The data were analyzed using FlowLogic™ Software (Inivai Technologies, Melbourne, Australia).

### 2.8. Statistical Analysis

Statistical analysis was performed using a two-way ANOVA for multiple comparisons. The analyses were conducted using GraphPad Prism 6 (GraphPad Holdings, San Diego, CA, USA). Significance is indicated in the figures as: * *p* < 0.05, ** *p* < 0.01, *** *p* < 0.001, **** *p* < 0.0001.

## 3. Results

### 3.1. Production and Characterization of C-Cys-Engineered Saporin (C-Sap)

In order to be produced, purified and subsequently conjugated to colloidal nanoparticles, a highly potent mature form of Sap (SAP-3; [25]) was genetically modified with the insertion of a Cys residue at the C-terminus and a His Tag at the N-terminus (Appendix A). Regarding the Cys insertion, it is worth noting that the conjugation strategy adopted for MYTS-Sap conjugation was selected for two main reasons: (1) disulfide bridges are selectively reduced in the cell cytoplasm, promoting the release of C-Sap in its site of action; (2) native Sap does not contain Cys residues, so the conjugation is highly specific and is not expected to affect the protein’s structural integrity.

The recombinant protein was expressed in *E. coli* (strain BL21(DE3)) and isolated from the soluble fraction by affinity chromatography. The presence of Sap and the purity of the fractions containing His-Tag proteins were checked by SDS–PAGE after Coomassie blue staining and WB analysis (Appendix A). The fractions containing most of the protein (lanes 3, 4 and 5 in Appendix A) were collected together and the buffer exchanged via centrifugal filtration. This last step is crucial because an inefficient elimination of imidazole employed for the protein elution significantly compromises the freeze–thaw stability of C-Sap. The purification protocol used here was designed to allow the obtainment of around 2.8 mg of C-Sap/L culture, similar to what was previously reported [26].

### 3.2. Synthesis and Characterization of Saporin-Loaded Nanoparticles (MYTS-Sap)

Twelve-nanometer iron oxide nanocrystals (Figure 1A) were coated with a biocompatible amphiphilic polymer (PMDA), allowing the obtainment of a highly stable NP dispersion with surface carboxylic functional groups useful for bioconjugation (Figure 1B,C). MYTS-Sap NPs were synthesized according to an optimized procedure derived from Fiandra et al., with some adjustments [12] Briefly, MYTS-PMDA nanoparticles were modified with 2,2-(ethylenedioxy)bis(ethylamine) (EDBE) to introduce amino functionalities that allowed us to incorporate thiol-reactive groups onto the PMDA surface by reaction with Sulfosuccinimidyl-6-(3′-(2-pyridyldithio)propionamido)hexanoate (sulfo-LC-SPDP), leading to the formation of MYTS-SPDP. The functionalization steps did not alter the colloidal stability (measured by hydrodynamic size; Figure 1B, Table 1). The surface negative charge, determined by zeta potential (Table 1), was only slightly decreased upon conjugation with Sap, thus maintaining the original colloidal stability due to electrostatic repulsion. MYTS-SPDP particles were incubated with C-Sap at different concentrations to achieve toxin bioconjugates (Figure 1C). We optimized the reaction conditions to obtain three different batches of MYTS-Sap having ~1, 2 or 4 C-Sap molecules per NP, respectively, with a conjugation efficiency of around 40% (see Table 2). As expected, upon increasing the number of C-Sap molecules attached to the NPs, the hydrodynamic size of the nanoconjugates was augmented (Table 1). Western blot analysis performed in reducing and non-reducing conditions confirmed the presence of C-Sap in all three nanoconjugates and specific conjugation through disulfide bridging (Appendix A). These three batches were designed to evaluate the effect of C-Sap concentration (i.e., the number of C-Sap molecules loaded on the NPs) on the cytotoxic activity of the nanoconjugates.

### 3.3. Antitumor Activity of MYTS-Sap

To assess the general cytotoxicity of MYTS-Sap, SK-BR-3 breast cancer cells were cultured and incubated with MYTS-SPDP and MYTS-Sap at increasing NP concentrations, i.e., 25, 37.5 and 50 µg/mL (Figure 2A). Although MYTS-SPDP nanoparticles were not toxic, MYTS-Sap particles exhibited dose-dependent activity. Considering the low protein uptake by cancer cells usually observed with the soluble toxin, we decided to increase the tested C-Sap concentration by up to four-fold compared to the highest amount loaded to the NPs (MYTS-Sap-3) (Appendix A). Nevertheless, even at the highest concentration tested, unconjugated C-Sap was much less cytotoxic compared to MYTS-Sap. Apparently, since no difference was found between MYTS-Sap-1, -2 and -3, the number of C-Sap molecules attached to the NPs does not affect the antitumoral activity of MYTS-Sap. These results suggest that to achieve an effective cytotoxic effect, the principal objective is to improve Saporin internalization and endosomal escape through optimized carriers. Figure 2B shows that while healthy fibroblasts (NIH-3T3 murine fibroblasts) are generally more sensitive to non-functionalized NPs (i.e., MYTS-SPDP), these cells were not affected by MYTS conjugation with C-Sap—either MYTS-Sap-1, -2 or -3—all of which have the same toxicity level as MYTS-SPDP. Notably, unconjugated C-Sap was nontoxic at the highest concentration. Overall, the data reported in Figure 2A,B are encouraging, as MYTS-Sap proved to be selective toward SK-BR-3 breast cancer in a dose-dependent manner.

To attempt to understand the reason for the selective MYTS-Sap activity against breast cancer cells, the uptake of MYTS-Sap in SK-BR-3 was compared with that in NIH-3T3 under the same experimental conditions. The extent of MYTS uptake was assessed by flow cytometry analysis using C-Sap labeled with Alexa Fluor™-647-NHS-ester. Figure 3A shows that SK-BR-3 exhibited a remarkably higher uptake of MYTS-Sap compared to NIH-3T3. The NP concentration was kept constant and the increasing intensity of fluorescence from MYTS-Sap-1 to MYTS-Sap-3 was only due to the different number of C-Sap molecules per NP. Indeed, Figure 3B shows the result obtained with MYTS-Sap normalized to the number of toxin molecules. Although the level of MYTS-Sap uptake in NIH-3T3 was significantly lower than in SK-BR-3, it was not negligible. Combining these data with the absence of toxicity in NIH-3T3 (Figure 2B), the results of the uptake experiment suggested that a very low number of internalized NPs were able to escape the endosomes and release Sap molecules into the cell cytoplasm. Furthermore, despite the fact that the protein detected in SK-BR-3 increased in MYTS-Sap-1, -2 and -3 according to the relative proportions of C-Sap/NP (Figure 3A), the lack of substantial differences previously observed in the MTT assay (Figure 2A) suggests that ligand density may somehow affect the efficiency of endosomal escape.

### 3.4. Mechanism of MYTS-Sap Cytotoxic Activity

To confirm that the cytotoxic activity of MYTS-Sap in cancer cells could be indeed attributed to the effect of C-Sap release in the cell cytoplasm and not to an unspecific toxicity of the nanoconjugate, we investigated two specific molecular pathways of C-Sap activity. We first assessed the apoptosis/necrosis profile by Annexin-V-FITC/PI double staining assay, and, secondly, we determined the toxin-mediated protein synthesis inhibition [1].

Figure 4 shows the results of the flow cytometry analysis performed on SK-BR-3 (Figure 4A) and NIH-3T3 (Figure 4B) and the relevant histograms reflecting the relative ratios of viable, necrotic and apoptotic cells (Figure 4C,D, respectively).

Both in SK-BR-3 and NIH-3T3, unconjugated C-Sap did not affect the apoptosis/necrosis rate compared to the untreated control, in accordance with the results of the MTT assay. In contrast, following incubation with MYTS-Sap in SK-BR-3, we observed a remarkable increase in the number of apoptotic cells and, to a much lower extent, also of necrotic cells. Confirming the data from the MTT assay, the entity of apoptotic effect was independent of the number of C-Sap molecules bound to the MYTS NPs (Figure 4C). Once again, MYTS-Sap nanoparticles were ineffective toward NIH-3T3 cells (Figure 4D).

A protein synthesis inhibition assay was performed to test the generally accepted main mechanisms of intoxication and cell death induced by Saporin. Figure 5A,C shows that MYTS-Sap nanoparticles were able to significantly decrease the amount of newly translated proteins in SK-BR-3 cancer cells. In contrast, no evidence of proteins synthesis inhibition was observed for NIH-3T3 control cells (Figure 5B,C), confirming the selective action of MYTS-Sap in cancer cells.

### 3.5. Dependence of MYTS-Sap Activity on Cancer Cell Lines

To explore the generalization of the cytotoxic activity of MYTS-Sap against cancer cells, MYTS-Sap were also tested in HeLa cervical cancer cells (Figure 6A) and in U87 glioblastoma cells (Figure 6B) through MTT assay. In all cancer cell cultures, dose-dependent cytotoxic activity of MYTS-Sap was observed. However, HeLa cells were much more sensitive to C-Sap nanoconjugates; thus, lower concentrations (in the range of 1–10 µg/mL) were required to investigate MYTS-Sap activity. Despite a reduction in cell viability being recorded in MYTS-SPDP-treated U87 cells, the effect displayed by Sap nanoconjugates was significantly higher. This is in contrast with what we observed with NIH-3T3, where no difference was found between bare and conjugated MYTS (Figure 2B).

## 4. Conclusions

In summary, we have investigated the impact of nanoconjugation of a highly cytotoxic RIP, namely, Saporin, on therapeutic toxin activity against cancer cells. Despite the efforts dedicated to the development of new therapeutic options based on RIPs for the treatment of aggressive neoplasms, their clinical application remains elusive. Among the various limitations that have been identified in their application in cancer therapy, fundamental concerns have been raised regarding Sap stability against degradation from proteases and its capability to reach tumor tissue, enter the cell membrane and escape the endosome to interfere with ribosome activity and inhibit protein synthesis. In this work, we focused on exploring the potential of conjugating Sap to colloidal NPs suitably coated with an amphiphilic polymer which have the capacity to promote endosomal escape [14]. We prepared three batches of Sap-conjugated NPs, namely, MYTS-Sap-1, MYTS-Sap-2 and MYTS-Sap-3, bearing, respectively, 1, 2 and 4 Sap molecules on average. The treatment of SK-BR-3 (breast), HeLa (cervical) and U87 (brain) cancer cells with the three batches of MYTS-Sap nanoparticles resulted in dose-dependent cytotoxic activity. Surprisingly, despite protein internalization increasing with the number of conjugated Sap/NPs, no difference between the three batches was apparently recorded in any of the activity experiments. In contrast, control NIH-3T3 healthy fibroblasts did not exhibit sensitivity to Sap nanoconjugates. Evidence of the actual involvement of Sap in the mechanism of action of MYTS-Sap toxicity was provided by an uptake experiment combined with an assessment of protein synthesis inhibition in SK-BR-3 cells. The same experiments performed in NIH-3T3 cells confirmed the selective proapoptotic activity of the protein delivered by MYTS NPs.

Notably, the structural features of MYTS promoted cellular internalization in cancer cells selectively and mediated an efficient endosomal escape and Sap release in the cytoplasm. This unpredicted behavior suggests the potential of MYTS-Sap as a cancer-selective drug delivery system even in the absence of conjugation with targeting ligands, which is favorable for clinical translation. Such a strong advantage will deserve further mechanistic investigations.

Overall, the results of our study implied that the newly designed MYTS-Sap nanoparticles were highly cytotoxic against cancer cells, with broad applicability, inducing apoptotic effects mediated by inhibition of translocation of tRNA in the ribosome complex, thus blocking protein synthesis. Remarkably, cytotoxic activity of MYTS-Sap was highly selective against cancer cells in vitro, preserving healthy fibroblasts from apoptosis induction. This was achieved without the insertion of any targeting agent, highlighting the potential of this nanoformulation. In addition, MYTS-Sap cytotoxicity proved to be independent of the amount of Sap conjugated to NPs but dependent (only) on the concentration of Sap nanoconjugates. This may have been due to the influence of ligand density on the extent to which Sap reached the cytoplasm as well as the achievement of a toxin activity plateau in tested conditions. Additional investigations are required to corroborate either of the two hypotheses. This preliminary study encourages the investigation of the use of colloidally stable Sap nanomedicines for the development of innovative treatment approaches against solid tumors.

## Figures and Tables

**Figure 1 pharmaceutics-14-01517-f001:**
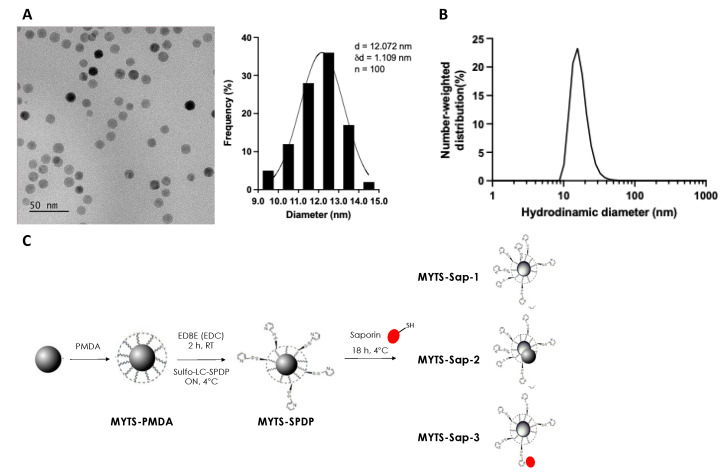
(**A**) TEM image of iron oxide cores. Scale bar: 50 nm; size distribution (*n* = 100). (**B**) Size distribution of MYTS-PMDA evaluated by DLS. (**C**) Schematic representation of MYTS-Sap synthesis reactions.

**Figure 2 pharmaceutics-14-01517-f002:**
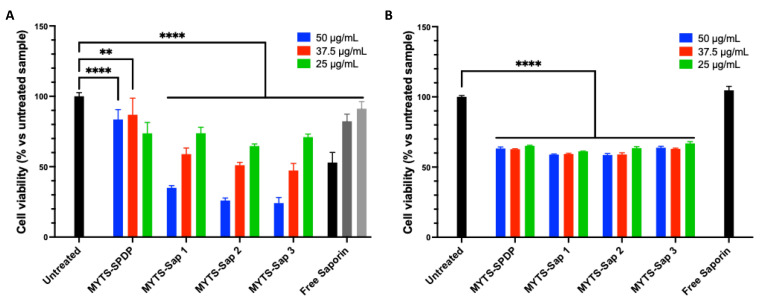
MTT assay performed incubating MYTS-SPDP, MYTS-Sap-1, MYTS-Sap-2 and MYTS-Sap-3 at concentrations of 25, 37.5 and 50 µg/mL with (**A**) SK-BR-3 cells and (**B**) for NIH-3T3 cells for 24 h. Corresponding Saporin concentrations (nM) are reported in Appendix A. For free Saporin samples, approximately 2× (light gray), 3× (dark grey) and 4× (black) the maximum Saporin concentrations tested with MYTS-containing samples (50 µg/mL MYTS-Sap-3, 111.1 nM) were used for SK-BR-3, and the maximum concentration was used for NIH-3T3. **** *p* < 0.0001, ** *p* < 0.01 vs. untreated (black).

**Figure 3 pharmaceutics-14-01517-f003:**
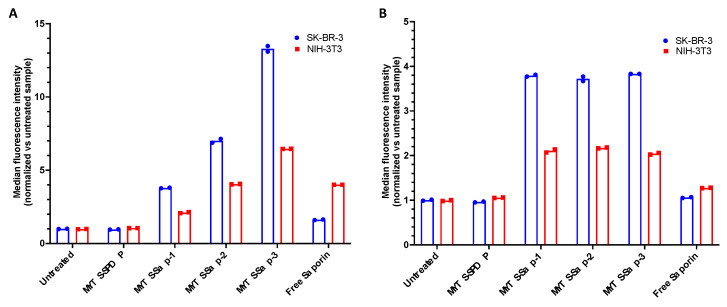
Cellular uptake of fluorescently labelled MYTS-Sap-1, MYTS-Sap-2 and MYTS-Sap-3 as evaluated by flow cytometry, with the incubation of an (**A**) equal MYTS concentration of 10 µg/mL or (**B**) an equivalent concentration of Saporin (5.4 nM) for 3 h. In both cases, free Saporin was used at the highest Saporin concentration tested (the one found in MYTS-Sap-3 equal to 22.2 nM and 5.4 nM, respectively).

**Figure 4 pharmaceutics-14-01517-f004:**
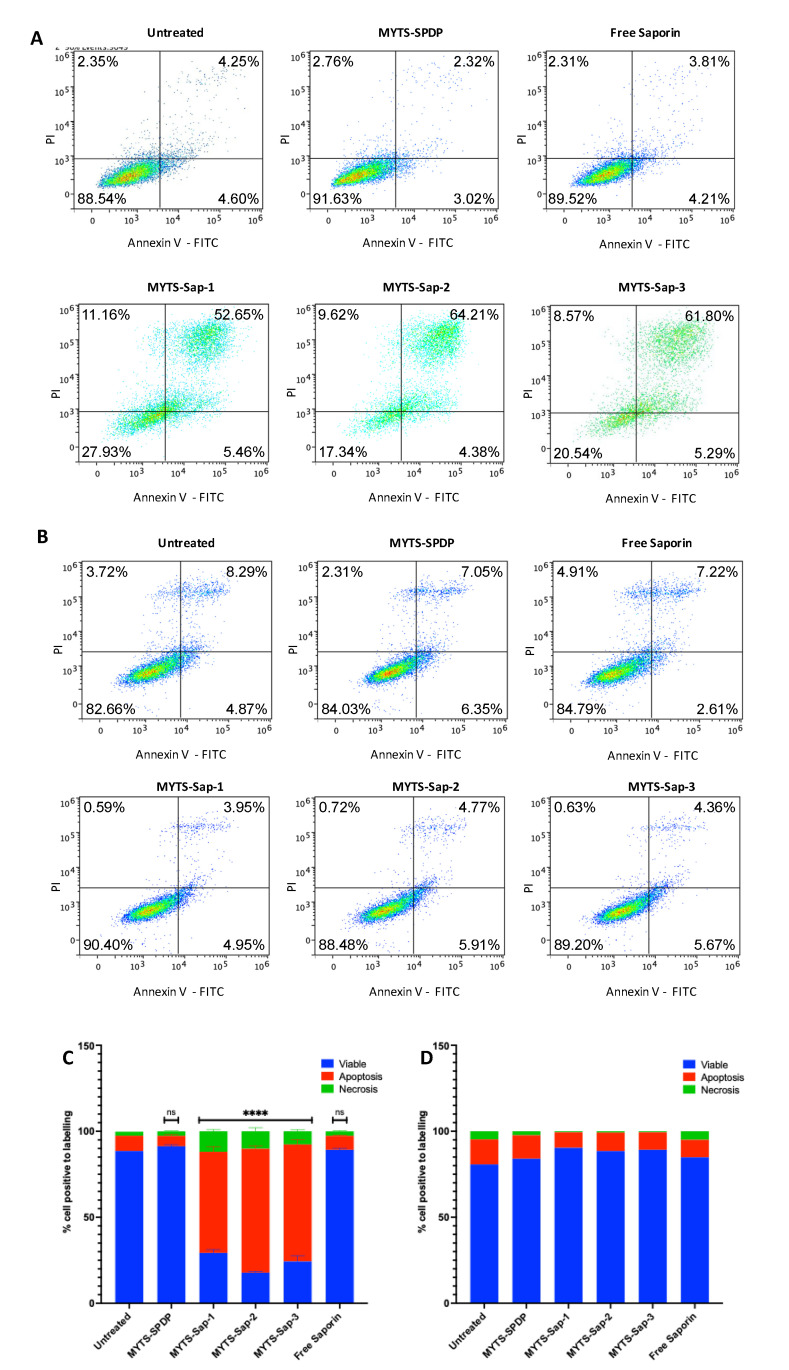
Annexin-V-FITC/PI double staining assay for the detection of apoptosis and/or necrosis performed on (**A**,**C**) SK-BR-3 cells and (**B**,**D**) NIH-3T3 cells, following 24 h of incubation with MYTS-SPDP, MYTS-Sap-1, MYTS-Sap-2 and MYTS-Sap-3 at concentrations of 75 µg/mL and free Saporin at the equivalent highest concentration found in the MYTS-treated samples (MYTS-Sap-3, approximately 166.6 nM). Viable: lower left quadrant; Apoptosis: lower + upper right quadrants; Necrosis: upper left quadrant. **** *p* < 0.0001; ns = non-significant vs. untreated.

**Figure 5 pharmaceutics-14-01517-f005:**
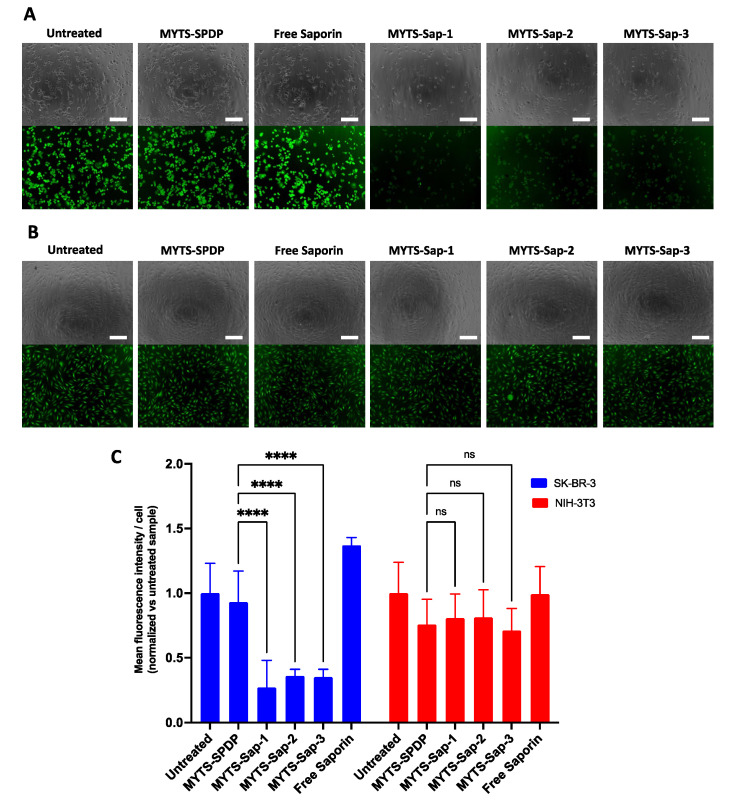
Protein synthesis inhibition assay performed on (**A**) SK-BR-3 cells and (**B**) NIH-3T3 cells, following 6 h incubation with MYTS-SPDP, MYTS-Sap-1, MYTS-Sap-2 and MYTS-Sap-3 at concentrations of 50 µg/mL and free Saporin and the equivalent highest concentration found in MYTS-treated samples (approximately 111.1 nM). Scale bars: 200 µm. (**C**) Quantification of fluorescence intensity/cell normalized against the untreated sample. **** *p* < 0.0001; ns = non-significant vs. MYTS-SPDP.

**Figure 6 pharmaceutics-14-01517-f006:**
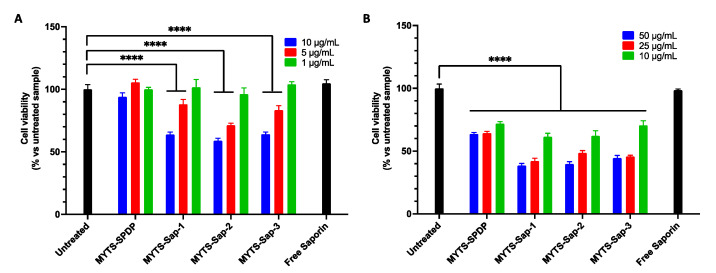
MTT assay performed incubating MYTS-SPDP, MYTS-Sap-1, MYTS-Sap-2 and MYTS-Sap-3 at concentrations of (**A**) 1, 5 and 10 µg/mL, with HeLa cells, and (**B**) 10, 25 and 50 µg/mL, with U87 cells, for 24 h. For free Saporin samples (black), the maximum Saporin concentration tested in MYTS-containing samples was used (10 µg/mL MYTS-Sap-3 for HeLa and 50 µg/mL MYTS-Sap-3 for U87). **** *p* < 0.0001 vs. untreated (black).

**Table 1 pharmaceutics-14-01517-t001:** Summary of DLS and zeta potential characterization of nanoparticles throughout the synthesis process. Data are expressed as the means of three measurements ± SDs.

Sample	Number-Weighted Hydrodynamic Diameter (nm)	Zeta Potential (mV)
MYTS-PMDA	17.2 ± 2.5	−46.0 ± 4.4
MYTS-SPDP	17.0 ± 4.6	−45.1 ± 3.2
MYTS-Sap-1	20.0 ± 1.5	−43.4 ± 2.9
MYTS-Sap-2	22.5 ± 6.8	−42.2 ± 0.7
MYTS-Sap-3	29.3 ± 4.7	−40.0 ± 0.9

**Table 2 pharmaceutics-14-01517-t002:** Quantification of Saporin conjugation in MYTS-Sap-1, MYTS-Sap-2 and MYTS-Sap-3. Conjugation efficiencies are expressed as the means of three independently synthesized batches ± SDs.

Sample	Reacted Saporin/MYTS (µg/mg)	Conjugation Efficiency (%)	Calculated Saporin/MYTS (Molar Ratio)
MYTS-Sap-1	50	37.4 ± 10.2	0.89
MYTS-Sap-2	100	37.0 ± 5.2	1.76
MYTS-Sap-3	200	38.6 ± 4.2	3.67

## Data Availability

Not applicable.

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
