# Peer review of "Saporin Toxin Delivered by Engineered Colloidal Nanoparticles Is Strongly Effective against Cancer Cells"

_pharmaceutics, 2022, doi:10.3390/pharmaceutics14071517_

Round 1

Reviewer 1 Report

Comments to the Authors:

The submitted paper concisely describes the use of Saporin toxin-loaded magnetic NPs as potential agents in anti-cancer therapies that were characterized on the cell lines. I include below several issues needed to be addressed in the manuscript before publication in Pharmaceutics

1. In the Introduction, the Authors used “MYTS” and “Sap” to describe magnetite nanoparticles and Saporin toxin. Can the Authors explain this choice (despite clear linkage to the previous papers)?

2. The Authors used magnetic NPs. Were there some reasons behind such a choice? If so, the Introduction and Discussion part should include a short note on that. 

3. In Table 1, the Authors included zeta potential results for the functionalized NPs. I did not find the comment on the obtained values. Is there good colloidal stability according to these results? Why did the addition of Saporin change the zeta potential value? How could it influence the behavior of these agents in the human cells during the ‘real’ treatment? 

4. The Authors concluded that Saporin toxin caused the toxicity of the MYTS NPs according to the comparison with the control cell line. This is a reasonable conclusion from the presented figures. However, it is commonly known that magnetic NPs possess their inherent toxicity as well. I wonder if such an effect was considered and if it should not be provided in the general discussion of the results. 

5. Generally, the novelty of this work is not clearly stated in the manuscript. It should be stressed in the Introduction and Conclusion section what the main message of the work is. 

The overall language of the manuscript is very understandable. However, careful proofreading of the whole text is required. There are some small technical issues and mistakes I found.
For instance:

a.   The sentence: One possible approach to overcome this issue is to exploit the excellent stability of Saporin to bioconjugation to deliver the toxin through a suitable nanocarrier that improves the intracellular release is probably too long and hardly clear. 

b.  Eq. (1-4) should be checked due to the technical criteria of the Journal (especially Eq. (4)).

c.       In the Conclusion, [9] reference is after the “.”. 

Author Response

  1. In the Introduction, the Authors used “MYTS” and “Sap” to describe magnetite nanoparticles and Saporin toxin. Can the Authors explain this choice (despite clear linkage to the previous papers)?

MYTS refers to the public name of the patented nanoparticles used in this work (patent: M. Colombo, F. Corsi, F. Granucci, D. Prosperi, I. Zanoni. “Nanoconstructs with pharmacological activity”. Patent number: WO2014013473-A1.)    

Sap is one of the common abbreviations of Saporin

  1. The Authors used magnetic NPs. Were there some reasons behind such a choice? If so, the Introduction and Discussion part should include a short note on that. 

The choice of MYTS nanoparticles is not directly related to their magnetic behavior, but to their specific and unique capacity to escape the endosomes, as already stated in the main text. For this reason, we did not point out their magnetic property because it is not relevant to the aim of the work.

  1. In Table 1, the Authors included zeta potential results for the functionalized NPs. I did not find the comment on the obtained values. Is there good colloidal stability according to these results? Why did the addition of Saporin change the zeta potential value? How could it influence the behavior of these agents in the human cells during the ‘real’ treatment? 

Thank you for highlighting this point that allowed  us to identify an oversight in the MYTS and MYTS-SPDP reported zeta potential data. Values were referred to measurements performed at 20 ul/ml concentration  in water whereas the other samples were analyzed at 30 ug/ml. We checked and corrected all the data summarized table 1 and commented on those.

  1. The Authors concluded that Saporin toxin caused the toxicity of the MYTS NPs according to the comparison with the control cell line. This is a reasonable conclusion from the presented figures. However, it is commonly known that magnetic NPs possess their inherent toxicity as well. I wonder if such an effect was considered and if it should not be provided in the general discussion of the results. 

One of main features of our MYTS nanoparticles is their very low intrinsic cytotoxicity at the concentrations tested in this work (very low iron concentrations). This was experimentally confirmed in this work resulting in a > 80% viability and negligible necrotic or apoptotic effect in absence of Sap (MYTS-SPDP).

  1. Generally, the novelty of this work is not clearly stated in the manuscript. It should be stressed in the Introduction and Conclusion section what the main message of the work is. 

We agree with the reviewer suggestion and substantially improved the discussion in the conclusion making the take home message more focused.

The overall language of the manuscript is very understandable. However, careful proofreading of the whole text is required. There are some small technical issues and mistakes I found. 

We thank the reviewer and check and correct the cumbersome sentences throughout the manuscript. 

Reviewer 2 Report

This work demonstrates the anti-cancer efficacy of amphiphilic polymer coated Saporin iron oxide nanoparticles. The conjugate may promote the endosomal escape and mediate cancer cell death by apoptosis, in comparison to healthy fibroblasts. I suggest the manuscript should be revised before further evaluation:

1) Since all nanoparticles are negatively charged (Table 1), how can they be effectively internalized in cells? Can the authors determine the cell uptake mechanism?

2) Figure 2, 3 and 6, label the significant differences among the experimental and control groups, as those exhibited in Figure 5c.

3) The cytometric plots in Figure 4 are illegible. The resolution of the graphs should be improved.

4) Add scale bars in Figure 5a.

5) Generally, the citations are not well updated. For example, in paragraph 3, Introduction section, some relevant works around RIPs toxin family should also be included.

Author Response

1) Since all nanoparticles are negatively charged (Table 1), how can they be effectively internalized in cells? Can the authors determine the cell uptake mechanism?

As previously demonstrated in several works from our group, MYTS nanoparticles exhibit strong capacity of being internalized in cancer cells exploiting the amphiphilic property of the coating polymer.

2) Figure 2, 3 and 6, label the significant differences among the experimental and control groups, as those exhibited in Figure 5c.

Done. Note that figure 3 has not reported the statistical analysis because it was tested in duplicate.

3) The cytometric plots in Figure 4 are illegible. The resolution of the graphs should be improved.

Done

4) Add scale bars in Figure 5a.

Done

5) Generally, the citations are not well updated. For example, in paragraph 3, Introduction section, some relevant works around RIPs toxin family should also be included.

Some new references have been added to properly support the text.
